# ILC3 GM-CSF production and mobilisation orchestrate acute intestinal inflammation

Claire Pearson[1,2†], Emily E Thornton[1,2†], Brent McKenzie[3‡], Anna-Lena Schaupp[1,2], Nicky Huskens[1,2], Thibault Griseri[1,2], Nathaniel West[1,2], Sim Tung[4], Benedict P Seddon[4§], Holm H Uhlig[2,5], Fiona Powrie[1,2*]

[1]Kennedy Institute of Rheumatology, University of Oxford, Oxford, United Kingdom; [2]Translational Gastroenterology Unit, Nuffield Department of Clinical Medicine, John Radcliffe Hospital, University of Oxford, Oxford, United Kingdom; [3]CSL Ltd., Parkville, Australia; [4]Division of Immune Cell Biology, National Institute for Medical Research, London, United Kingdom; [5]Department of Paediatrics, John Radcliffe Hospital, University of Oxford, Oxford, United Kingdom

*For correspondence: fiona. powrie@kennedy.ox.ac.uk

†These authors contributed equally to this work

Present address: ‡Genentech, Inc., South San Francisco, United States; §Institute of Immunity and Transplantation, Division of Infection and Immunity, University College London, London, United Kingdom

**Abstract** Innate lymphoid cells (ILCs) contribute to host defence and tissue repair but can induce immunopathology. Recent work has revealed tissue-specific roles for ILCs; however, the question of how a small population has large effects on immune homeostasis remains unclear. We identify two mechanisms that ILC3s utilise to exert their effects within intestinal tissue. ILC-driven colitis depends on production of granulocyte macrophage-colony stimulating factor (GM-CSF), which recruits and maintains intestinal inflammatory monocytes. ILCs present in the intestine also enter and exit cryptopatches in a highly dynamic process. During colitis, ILC3s mobilize from cryptopatches, a process that can be inhibited by blocking GM-CSF, and mobilization precedes inflammatory foci elsewhere in the tissue. Together these data identify the IL-23R/GM-CSF axis within ILC3 as a key control point in the accumulation of innate effector cells in the intestine and in the spatio-temporal dynamics of ILCs in the intestinal inflammatory response.

## Introduction

Innate lymphoid cells (ILCs) are a recently defined family of evolutionarily ancient cells involved in many facets of host defence. As with conventional Th1, Th2 and Th17 T cells, ILCs can be functionally classified based on the expression of transcription factors and associated signature cytokines (*Spits et al., 2013*). ILC1 are defined by Th1-like and ILC2 by Th2-like cytokine responses, whereas ILC3 express RORγt, and can secrete IL-17 and/or IL-22 upon activation. The ILC3 family includes the prototypic foetal lymphoid tissue inducer (LTi) cells that play a non-redundant role in lymphoid tissue development (*Mebius et al., 1997*) via lymphotoxin-dependent interactions with stromal cells. This pathway is required not only for lymph node development but also for organised lymphoid structures in the gut such as dendritic cell (DC) and ILC containing cryptopatches (CP) (*Eberl and Sawa, 2010*), B cell, DC and ILC containing isolated lymphoid follicles (ILFs) (*Tsuji et al., 2008*), and small intestinal Peyer's Patches (PP) that have a very similar structure to lymph nodes (*Cornes, 1965*; *Mebius et al., 1997*). Postnatally, the LTi population can also make IL-17 and IL-22 (*Cupedo et al., 2009*) and contribute to host defence against pathogens, particularly in the gut (*Sonnenberg et al., 2011*).

The multi-functional roles of ILCs in disease development and pathogenesis (*Buonocore et al., 2010*), as well as host defence (*Moro et al., 2010*; *Neill et al., 2010*; *Sonnenberg et al., 2011*) and repair (*Monticelli et al., 2011*) have been the focus of much interest. Polyfunctional ILCs have also been described that do not fall into distinct ILC1 or ILC3 phenotypes but express both ILC1 and

**eLife digest** Crohn's disease and ulcerative colitis are diseases in which the body's own immune system causes inflammation of the large intestine. These autoimmune diseases can be severely debilitating and difficult to treat. However an improved understanding of the factors that contribute to the intestinal inflammation may lead to new and effective treatments.

Immune cells called innate lymphoid cells were discovered recently, and shown quickly to play a role in host defense, tissue repair and inflammation regulation. Several groups of innate lymphoid cells are now known; each group is characterized by the genes that control the cell's development and the small proteins (called cytokines) that the cells release. One group of innate lymphoid cells, the ILC3s, are generally found in the intestinal tract, albeit in small numbers. Given that innate lymphoid cells are known to manage inflammatory responses, it is possible that ILC3s contribute to intestinal inflammation. However, it remains unclear how such a small population of cells could so dramatically inflame the gut.

Pearson et al. now reveal two mechanisms that these innate lymphoid cells use to amplify the inflammatory response and exacerbate intestinal inflammation. First, in both mice and humans, ILC3s were found to be a key source of a cytokine called GM-CSF, which recruits additional immune cells that further promote intestinal inflammation. Secondly, while ILC3s were traditionally regarded as immobile immune cells, Pearson et al. discovered that these cells can move within the intestinal tissue and mobilize from their starting points within this tissue if they are activated. These two mechanisms could explain how ILC3s can trigger inflammation that occurs throughout the gut.

The experiments suggest that blocking production of the GM-CSF cytokine or altering ILC3 movement or activity may help reduce intestinal inflammation. However, the use of GM-CSF blocking drugs to protect against colitis and similar conditions could be problematic, because GM-CSF also plays an important protective role in the intestines. Nevertheless, clinical trials are underway to investigate the use of anti-GM-CSF drugs to treat other inflammatory conditions (such as rheumatoid arthritis). These studies could offer insight into whether these drugs provide relief to trial participants who suffer from intestinal inflammation as well.

ILC3 lineage defining transcription factors Tbet and RORγt and secrete multiple cytokines such IL-17A, IL-22 as well as IFNγ (*Buonocore et al., 2010*).

We previously identified a critical role for a phenotypically distinct population of IL-23R$^+$ RORγt$^+$ CD4$^-$ NKp46$^-$ ILCs in the development of innate intestinal inflammation in *Rag*$^{-/-}$ mice following *Helicobacter hepaticus* infection or αCD40 stimulation (*Buonocore et al., 2010*). Similar ILC populations were enriched in the colonic mucosa of patients with inflammatory bowel disease (IBD) (*Geremia et al., 2011*), implicating IL-23-responsive, RORγt expressing ILCs in the pathogenesis of inflammatory gut disease in mice and humans. However, it remains unclear how ILCs, that are numerically sparse in vivo can initiate inflammatory processes that lead to colitis.

Despite advances in understanding of the functions of ILCs, little is known about their location in tissue at different stages of the inflammatory response, and how putative structural and cytokine-mediated functions are co-ordinated. Since its description in 2006 (*Uhlig et al., 2006*), the induction of colitis by injecting agonistic anti-CD40 antibody has become an important tool to assess ILC-driven acute colitis (*Buonocore et al., 2010*; *Vonarbourg et al., 2010*; *Fuchs et al., 2013*; *Kim et al., 2013*; *Song et al., 2015*). By contrast with other models, anti-CD40 induced colitis follows discrete phases at well-defined time points following initiation, offering the opportunity to probe the role of leukocytes in the development and amplification of the inflammatory response. Experiments have demonstrated that intestinal inflammation was mediated via Thy1$^+$ ILCs in a *rorc* dependent manner, making it an ideal system to study how ILCs contribute to pathogenesis (*Buonocore et al., 2010*). A recent study investigating potential biomarkers for anti-IL-23 therapy described similar changes in the colons of both anti-CD40-treated mice and patients with active Crohn's disease (*Cayatte et al., 2012*).

Many recent publications have focused on the specific functions of ILC subsets within effector sites, and the location of ILCs has been proposed to contribute to their ability to affect systemic cytokine levels (*Nussbaum et al., 2013*). Despite histological and flow cytometry data

demonstrating the presence of ILCs within lymphoid structures in the gut (*Eberl and Sawa, 2010*), it isn't clear whether they function as sedentary, cytokine producing cells or play a more active role in cell interactions and organization. In vivo microscopy is a tool that provides an opportunity to look at the behaviour of ILCs within the tissue. By combining anti-CD40 stimulation with intra-vital microscopy we are able to reliably track cellular changes at discrete phases of disease and capture cell movement at key timepoints.

Our results show two novel mechanisms through which the small number of ILCs found in vivo orchestrate the intestinal inflammatory response. IL-23-driven GM-CSF production by ILC3s is critical for the development of colitis, and ILCs mobilise from cryptopatches after activation in a GM-CSF-dependent manner. Both of these behaviours likely contribute to the ability of ILCs to coordinate the immune response in the gut. Initiation and perpetuation of disease occur in distinct anatomical compartments, indicating both a temporal and spatial switch of ILC function during inflammatory conditions.

## Results

### GM-CSF is a critical cytokine mediator in the pathogenesis of innate colitis

Anti-CD40 induced colitis is dependent on a RORγt/IL-23 axis but key downstream cytokines are less well understood (*Uhlig et al., 2006*; *Buonocore et al., 2010*). As IL-17 and IL-22 are major downstream effectors of the IL-23 signalling axis (*Zheng et al., 2007*; *McGeachy et al., 2009*) we first investigated their role in anti-CD40 colitis. However, blockade of IL-17A failed to modify anti-CD40-induced systemic or intestinal disease (*Figure 1A,B*), indicating that IL-17A is dispensable for development of acute colitis in this model. Blocking the closely related molecule IL-17F also failed to modify disease (*Figure 1—figure supplement 1*).

Like IL-17 blockade, treatment with a neutralising anti-IL-22 mAb had no effect on systemic disease or colitis in the proximal colon (*Figure 1A&B*), indicating that in our facility, IL-22 is also redundant for disease induction in this model. However, blockade of IL-17A and anti-IL-22 reduced intestinal inflammation but not systemic disease indicating redundant roles for IL-17A and IL-22 in the intestinal inflammatory response (*Figure 1—figure supplement 1*).

As we have described an important role for GM-CSF in IL-23 driven T cell dependent colitis we next investigated the role of GM-CSF in innate colitis models. Strikingly, administration of an anti-GM-CSF blocking mAb reduced both weight loss (*Figure 1C*) and severity of colitis (*Figure 1D*). Indeed amelioration of colitis with GM-CSF blockade was similar to that observed with anti-IL-23R mAb treatment. Similar effects of GM-CSF blockade were also observed in bacteria-induced innate colitis following *Helicobacter hepaticus* infection of 129SvEv *Rag2*$^{-/-}$ mice (*Figure1—figure supplement 2*) supporting a pivotal role for this cytokine in both T cell dependent and innate colitis. GM-CSF has been shown to promote CNS inflammation through effects on inflammatory monocytes (*Croxford et al., 2015*). Consistent with this and in line with a recent report (*Song et al., 2015*) GM-CSF blockade led to a marked reduction in the number of inflammatory monocytes in the colon of anti-CD40 treated mice (*Figure 1E*). The number of neutrophils and eosinophils also decreased, but this was not significant. These changes were accompanied by an increase in both the percentage and total number of CD11b$^-$ CD103$^+$ dendritic cells (*Figure 1E*).

### ILCs are an important source of GM-CSF in mouse and human

We next investigated GM-CSF expression by flow cytometry to identify the cellular source in intestinal inflammation. Only a small proportion of epithelial, stromal or lineage positive (CD11b, CD11c, Gr-1, B220, CD49b) cells expressed GM-CSF (*Figure 2A*). The majority of these lineage positive GM-CSF producers were NK cells, and they expressed a lower amount of GM-CSF than ILCs (*Figure 2—figure supplement 1A & B*). However, using Rag and IL-15 receptor double deficient mice that lack NK cells (*Lodolce et al., 1998*), we found, as previously reported (*Vonarbourg et al., 2010*), that NK cells were not required for the development of anti-CD40 mediated colitis (*Figure 2—figure supplement 1C–F*). By contrast, a significant proportion of ILCs were capable of producing GM-CSF, even in the healthy intestine, and this proportion increased during colitis

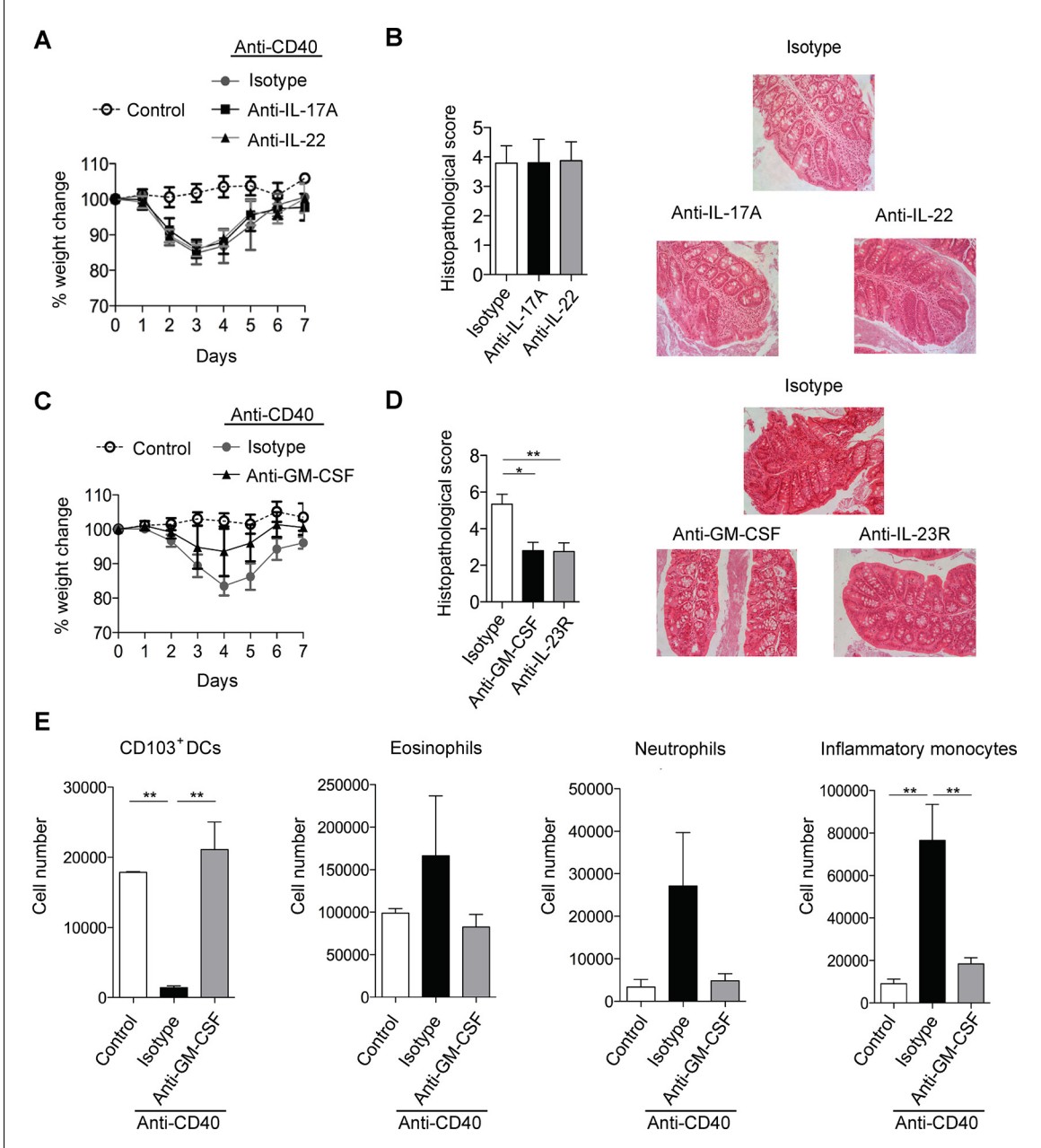

**Figure 1.** GM-CSF is a critical cytokine mediator of ILC-driven colitis. (**A**) Weight loss and (**B**) proximal colon histopathology scores in untreated B6*Rag1*[-/-] mice (control, n=3) or mice injected with anti-CD40 and treated with blocking antibody to IL-17A (n=5), IL-22 (n=7) or isotype (n=8) for 7 days. Representative photomicrographs of H&E stained proximal colon sections are shown. (**C**) Weight loss and (**D**) proximal colon histopathology scores in untreated B6*Rag1*[-/-] mice (control, n=8) or mice injected with anti-CD40 and treated with blocking antibody to GM-CSF (n=7), blocking antibody to IL-23R (n=5) or isotype (n=6) for 7 days. Representative photomicrographs of H&E stained proximal colon sections are shown. (**E**) Number of innate immune cell populations in cLP at three days following anti-CD40 injection. Data are shown as means and SEM. Results are representative of n=2–5 independent experiments. *, $p < 0.05$, **, $p < 0.01$, One-way ANOVA with Bonferroni's post test. (*Figure 1—figure supplement 1*) shows no effect of IL-17A and IL-22 double blockade and IL-17F blockade on systemic disease or colitis. (*Figure 1—figure supplement 2*) shows improved systemic and intestinal disease with anti-GM-CSF treatment in *Helicobacter hepaticus* driven innate colitis.

The following figure supplements are available for figure 1:

**Figure supplement 1.** IL-17A and IL-22 combination blockade or anti-IL-17F does not protect from anti-CD40 mediated colitis.

**Figure supplement 2.** *Helicobacter hepaticus* driven innate colitis depends on GM-CSF.

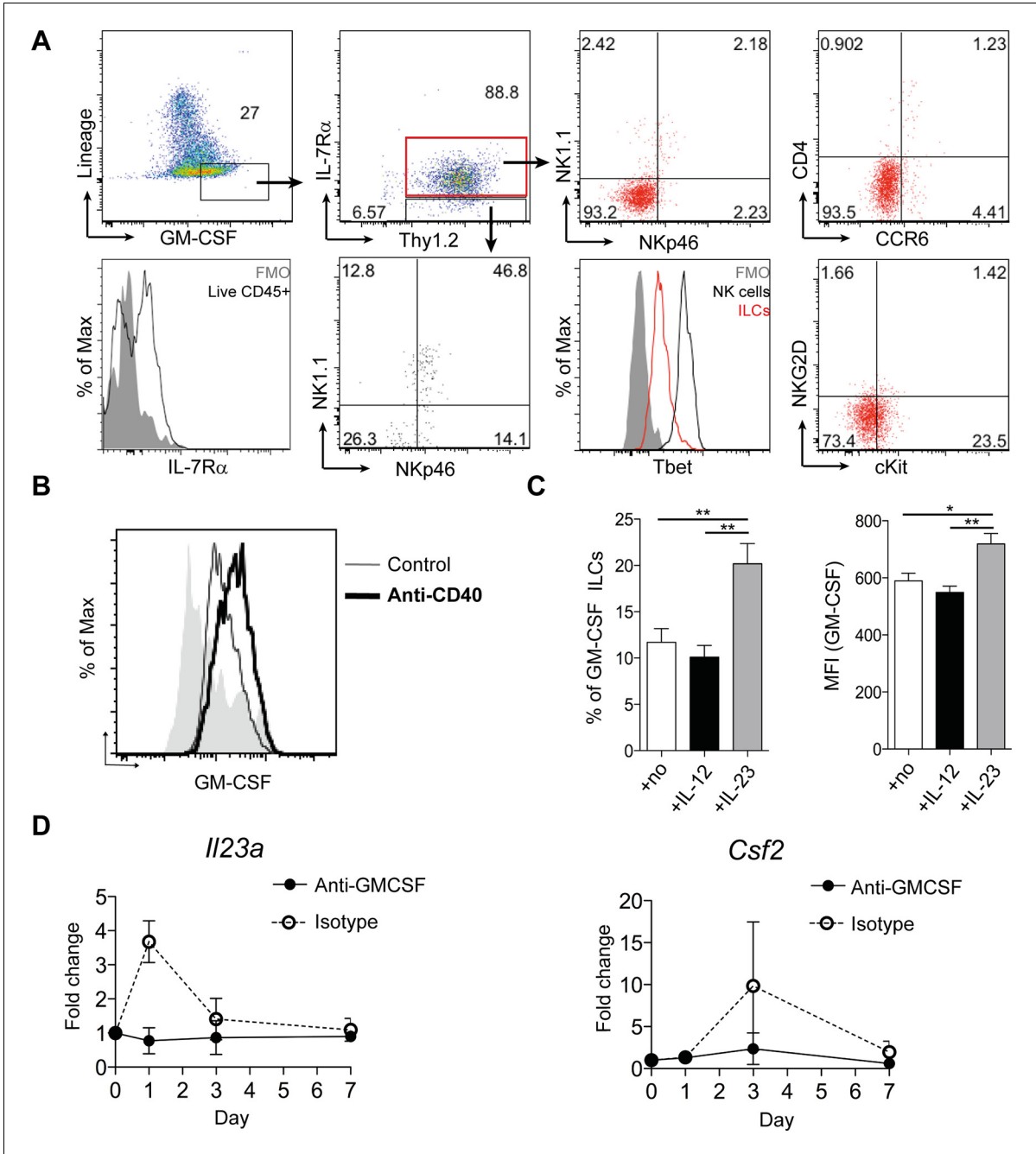

**Figure 2.** ILCs are a major source of GM-CSF. (**A**) Representative flow cytometric analysis of ILC populations at day 3 following anti-CD40 treatment. Surface marker and GM-CSF expression is shown following 3 hr stimulation with PMA, ionomycin, monensin, and brefeldin A (n=6). Cells are gated on a live cell gate with doublets excluded. GM-CSF expression gating is based on an FMO control. Red marks ILCs and black marks NK cells. Lower left panel shows IL-7Rα staining in live CD45[+] cells compared with FMO control. (**B**) Representative flow cytometric analysis showing GM-CSF production for ILC populations in untreated or anti-CD40 treated mice. The solid grey histogram represents the isotype control. Following doublet exclusion, cells are gated on a live cell gate, then gated on lineage-, CD45[+] Thy1.2[+] IL-7Rα[+] cells. (**C**) Percent and median fluorescence intensity of GM-CSF expression in CD4[-] NKp46[-] ILCs isolated from mice 7d following anti-CD40 injection. Total cLP cells were stimulated over night with IL-12, IL-23 or in medium alone, followed by 3 hr stimulation with PMA, ionomycin, monensin and brefeldin A (n=7). Results are representative of 2–3 independent experiments. (**D**) mRNA expression of *Il23a* and *Csf2* in proximal colonic lamina propria at various time points following anti-CD40 injection. Results are shown as fold change in target gene relative to *hprt* compared with day 0 uninjected control mice. Data are shown as means and SEM. Results are pooled from 2 independent experiments with n=4–6 mice per group per experiment. *, $p<0.05$, **, $p<0.01$, one-way ANOVA with Bonferroni's post test. (***Figure 2—figure supplement 1***) shows analysis of lack of NK cell contribution to colitis.

*Figure 2 continued on next page*

*Figure 2 continued*

The following figure supplement is available for figure 2:

**Figure supplement 1.** NK cells are not required for anti-CD40 induced colitis.

(*Figure 2B*). Consistent with an IL-23-dependent ILC3 phenotype, GM-CSF[+] ILCs were CCR6[-], cKit[-] and NKR[-], and expressed only low amounts of T-bet (*Figure 2A*).

We next investigated the cytokine milieu that regulates GM-CSF production by ILCs in vivo. To determine whether IL-23 signalling induces GM-CSF, we cultured total colonic lamina propria cells overnight with IL-12 or IL-23 and measured GM-CSF production in the cultures. IL-12 had no effect on either the proportion of GM-CSF-expressing ILCs or the amount of GM-CSF per cell (*Figure 2C*). By contrast and contrary to a previous study using sorted ILC3s (*Mortha et al., 2014*), culture with IL-23 increased both the proportion of GM-CSF expressing ILCs as well as GM-CSF production per cell (*Figure 2C*).

Analysis of *Il23a* mRNA expression within colonic tissue revealed a peak in expression at day 1 following anti-CD40 treatment that rapidly declined but was followed by increased expression of *Csf2*, the gene for GM-CSF, at day 3 (*Figure 2D*). The increase in both *Il23a* and *Csf2* was abrogated in mice treated with a blocking antibody to GM-CSF, indicating that GM-CSF may be required to maintain and amplify expression of *Il23a*.

To assess whether GM-CSF production may also be relevant in human IBD, we first analyzed GM-CSF production from blood ILCs. PMA and ionomycin stimulated blood ILC3s produced GM-CSF with very little production from other ILC subsets (*Figure 3A&B*, gating strategy in *Figure 3—figure supplement 1*). Indeed, a greater proportion of ILCs produced GM-CSF than did T cells from the same healthy donors (*Figure 3C*). ILCs secreting GM-CSF are further enriched in the colon (*Figure 3D*). Critically, the proportion of blood ILCs capable of GM-CSF production along with other disease-associated cytokines such as IFNγ and TNFα was greater in IBD patients than in healthy controls (*Figure 3E*).

Analysis of a publically available dataset of colon tissue from healthy controls and IBD patients showed a significant increase in *CSF2* gene expression in the colons of patients with either Crohn's disease or ulcerative colitis compared with controls (*Figure 3F*). This was confirmed by gene expression analysis of colon biopsies from the Oxford cohort of IBD patients (*Figure 3G*). CSF2 expression was also higher in uninflamed biopsies from IBD patients compared with controls. There was significantly greater expression again in lesional areas in active disease, suggesting that GM-CSF expression correlates with, and may be a driver of, inflammation in IBD.

## ILCs are present within cryptopatches but mobilize during colitis induction

To date attention has focussed on the capacity of ILCs in tissue to produce immune modulatory cytokines with little emphasis on how ILC positioning within a tissue may impact on functional outcomes. Under non-inflammatory conditions, ILCs are found within lymphoid structures in the colon (*Eberl and Sawa, 2010*), and in *Rag*-deficient hosts these structures are limited to cryptopatches as there are no B cells to form ILFs. Under homeostatic conditions ILCs primarily reside within lymphoid aggregate cryptopatch (CP) structures (*Figure 4A*) and are found only rarely within the non-inflamed lamina propria. Within CP, RORγt[+] and IL-7Rα[+] ILC3s are present at high density surrounded by CD11c and MHCII expressing cells. Some of these ILCs within CP also express CD4, suggesting that some but not all are classical LTi cells that are known to be involved in lymphoid organogenesis (*Figure 4A*). In addition to RORγt expression, ILCs are known to express IL-23R (*Buonocore et al., 2010*). FACS analysis of *Rag*-deficient mice that express GFP under control of the *Il23r* demonstrates that these mice can be used to study ILC behaviour in vivo (*Figure 4B*). Two-photon imaging of *Il23r*[gfp/+] *Rag*[-/-] mice shows ILCs (green) present within a cryptopatch (*Figure 4C*, and *Video 1*). Quantification of ILC localization in 3D indicates that greater than 90% of ILCs are present within cryptopatches under steady-state conditions (*Figure 4D*). This suggests that IL-23R[+] ILCs are perfectly positioned in close proximity to a high density of CD11c[+] DCs that can be activated by colitogenic stimuli. Indeed *Il23r* mRNA increased in the colon 6 hr following anti-CD40 injection indicating that the first ILC changes occur in the hours following CD40 stimulation (*Figure 4—figure supplement 1A*).

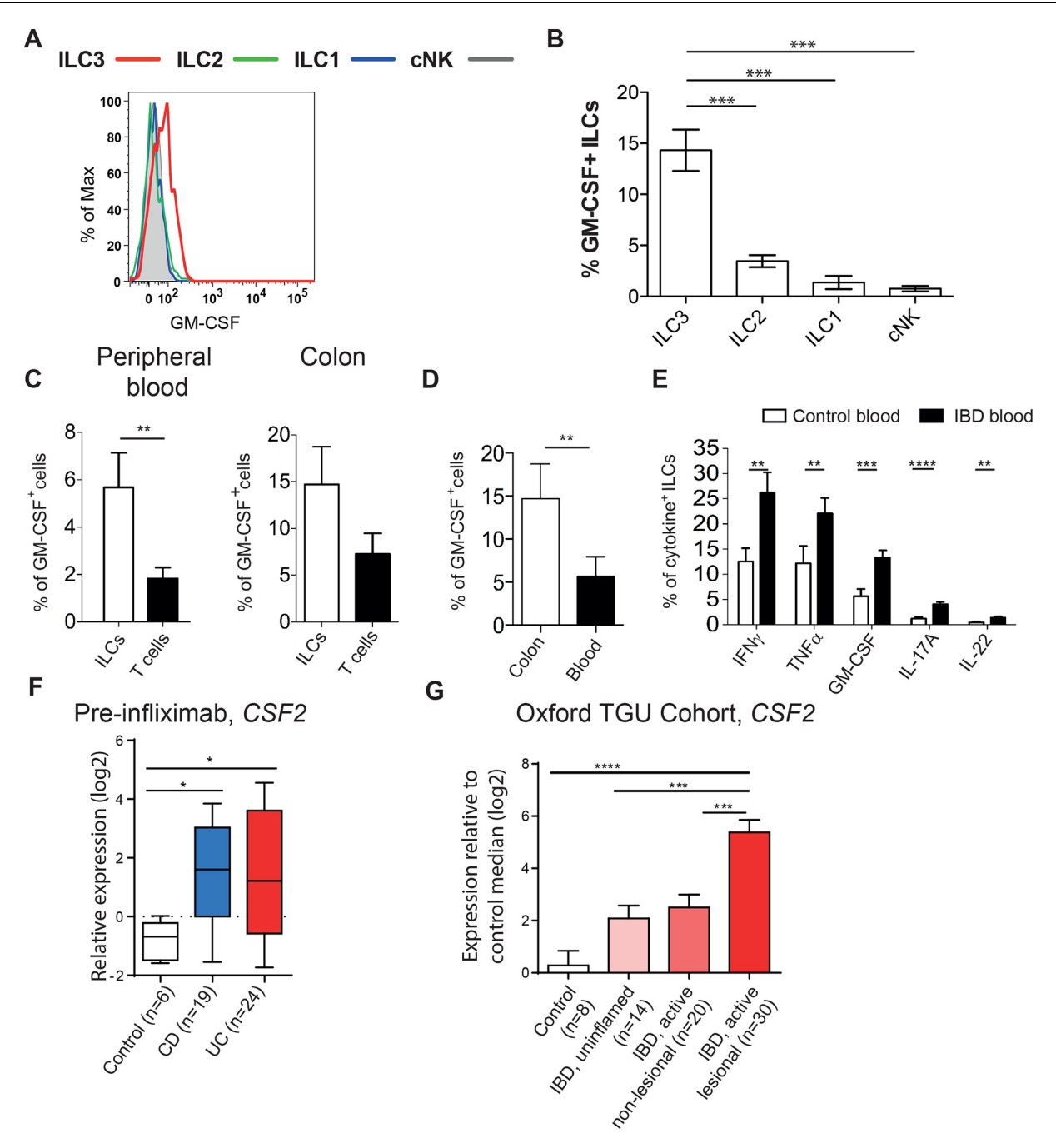

**Figure 3.** Human ILCs are a source of GM-CSF, which increases in IBD. (**A**) Representative flow cytometric analysis of GM-CSF in human blood ILC subsets (following doublet exclusion and gated on live cells, ILCs are lineage⁻. ILC1 are IL-7Rα⁺ cKit⁻ CRTH2⁻, ILC2 are IL-7Rα⁺ CRTH2⁺, ILC3 are IL-7Rα⁺ cKit⁺ CRTH2⁻. cNK cells are CD56⁺ IL-7Rα⁻ CD45RO⁻) stimulated with PMA, ionomycin and brefeldin A for 4 hr. (**B**) Quantification of percent of GM-CSF⁺ blood ILC populations (n=3–6). (**C**) Percent of human blood and colon ILCs and T cells expressing GM-CSF following stimulation with PMA, ionomycin and brefeldin A for 4 hr (n=5). (**D**) Comparison of ILCs in blood and colon expressing GM-CSF (n=5). (**E**) Percent of ILCs expressing pro-inflammatory cytokines in the blood of control and IBD patients following stimulation with PMA, ionomycin and brefeldin A for 4 hr (n=10–15). Results are representative of n=2–5 independent experiments. *, $p<0.05$, **, $p<0.01$, ***, $p<0.001$, ***, $p<0.0001$ one-way ANOVA with Bonferroni's post test. (**F**) Relative expression of *CSF2* from the publicly available Leuven cohort (GSE16879) in control, Crohn's disease, and ulcerative colitis patients before infliximab treatment. (**G**) Relative expression of *CSF2* in the Oxford IBD cohort. Data for individual genes were normalized to the median value of healthy control patients, converted to log₂ ratios, and analyzed by one-way ANOVA with Tukey's multiple comparisons test (data were found to be normally distributed using the D'Agostino and Pearson omnibus normality test).

*Figure 3 continued on next page*

*Figure 3 continued*

The following figure supplement is available for figure 3:

**Figure supplement 1.** Gating strategy for human ILC subsets.

We further examined the changes in IL-23R expression following anti-CD40 stimulation, and found a decrease in IL-23R⁺ ILCs at day 3 (*Figure 4—figure supplement 1B&C*), which may indicate a change in ILC phenotype. In support of this hypothesis, analysis of RORγt protein revealed a transient decrease in the proportion of ILCs expressing it 3 days following anti-CD40 stimulation (*Figure 4—figure supplement 1D&E*), similar to a previous report (*Vonarbourg et al., 2010*). Whilst these data suggested that ILCs are reacting to changes in the environment, it remained unclear how this leads to the downstream inflammatory cascade.

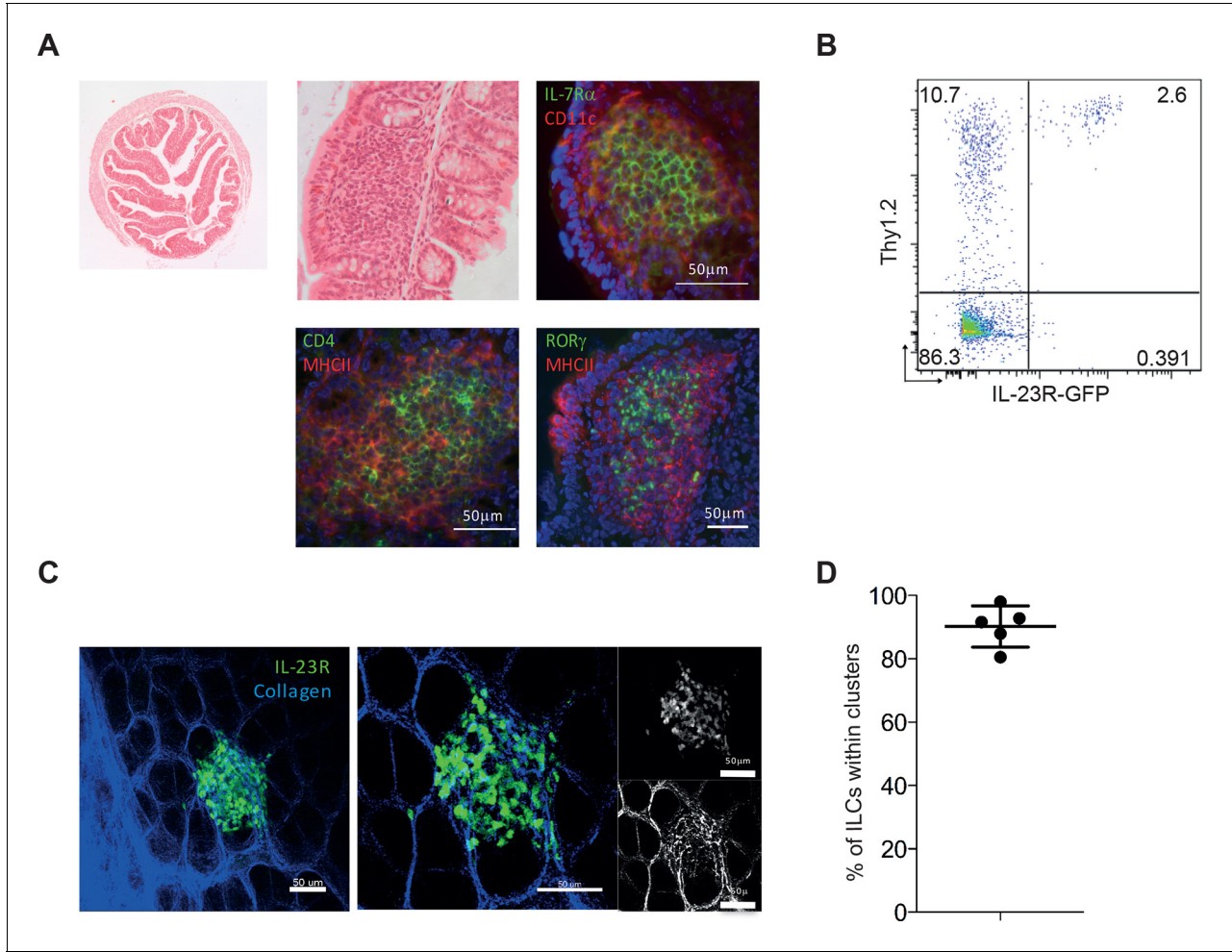

**Figure 4.** IL-23R marks ILCs that are present in cryptopatches within the gut. (**A**) Representative H&E and immunofluorescence staining of cryptopatches in transverse proximal colon sections in B6*Rag1*⁻/⁻ mice from the steady state at 2.5x and 20x magnification. (**B**) Flow cytometry staining of Thy1.2 in the colon LPLs of steady-state *Il23r*ᵍᶠᵖ/⁺ *Rag1*⁻/⁻. (**C**) Representative image of intact tissue explant of proximal colon from *Il23r*ᵍᶠᵖ/⁺ *Rag1*⁻/⁻ mice from the steady state. Left and middle panels show IL-23R (green) and collagen (blue). Right panel shows IL-23R alone (top) and collagen alone (bottom). (**D**) Quantification of steady-state *Il23r*ᵍᶠᵖ/⁺ *Rag1*⁻/⁻ ILCs in clusters within the proximal colon from explant imaging. (*Figure 4—figure supplement 1*) shows modulation of RORγt and IL-23R after anti-CD40 treatment. Results are representative of 3-4 independent experiments.

The following figure supplement is available for figure 4:

**Figure supplement 1.** RORγt and IL23R expression are affected by anti-CD40 treatment.

It is possible that ILCs that are resident within lymphoid structures produce cytokines locally to carry out their effects. Alternatively, ILCs may be able to mobilize into the tissue to coordinate the immune response. To understand which mechanism ILCs utilize, we adopted the strategy of McDole et al (*McDole et al., 2012*) used to study leukocytes in the gut. In the steady-state, ILCs were much more motile than expected. While little motility could be observed within the cryptopatch, ILCs can be observed to enter and exit the cluster near the basal collagen layer (*Figure 5A* and *Videos 2* and *3*). Interestingly, the cryptopatch did not have an obvious point of entry or exit, implying that trafficking into and out of the structure may be governed by mechanisms differ-

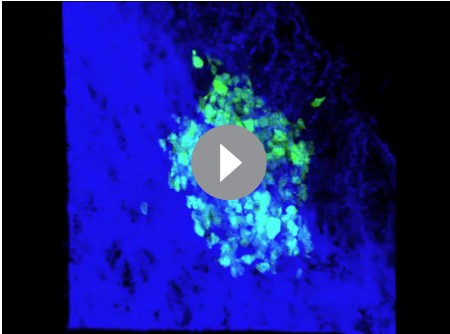

**Video 1.** 3D rotation of *Il23r*$^{gfp/+}$ *Rag1*$^{-/-}$ ILCs (green) within the collagen matrix of the gut (blue).

ent to that of the lymph node. By gating on motile cells outside the cluster, we could see that the average velocity of motile ILCs (approximately 9 μm/min) is grossly similar to that of other lymphocytes and while cell velocity decreases after treatment with anti-CD40 (*Figure 5B*), this difference does not appear to affect overall cell trafficking. To determine whether ILC trafficking changed in colitis, we plotted the displacements of ILCs in steady-state and after anti-CD40 treatment (*Figure 5C*, *Video 4*, and *Video 5*). While no gross changes appeared in track displacements, the overall movement of cells shifted. In steady-state, roughly equal numbers of cells move toward and away from the cluster; however, just 4–6 hr after anti-CD40 treatment, a greater proportion of cells exit the tissue and a smaller proportion enter (*Figure 5D*). The ratio of entering to exiting cells indicates a significant skewing toward egress from cryptopatches (*Figure 5E*). To assess whether GM-CSF may be playing a role in ILC mobilization, we treated mice with anti-GM-CSF 24 hr before imaging. The shift toward exit that is normally observed after anti-CD40 treatment was strikingly reversed when mice were treated with a GM-CSF blocking antibody 24 hr before imaging (*Figure 5D and E*, *Video 6*). It is possible that many of these cells move through blood vessels, but by labelling the blood vessels with texas-red dextran, ILCs appeared to be adjacent to blood vessels, not within them (*Figure 5F*). This indicates that ILCs may be mobilizing into the adjacent tissue from cryptopatches.

## ILC movement precedes tissue reorganization

The initial characterisation of the anti-CD40 model identified an influx of myeloid cells, particularly CD40$^+$ MHC-II$^+$ CD11c$^{hi}$ DCs, in specific inflammatory foci within the tip of the villi of the proximal colon during colitis (*Uhlig et al., 2006*). We hypothesised that development of colitis is driven by

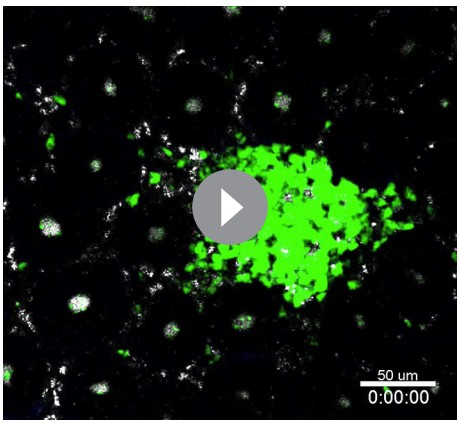

**Video 2.** Timelapse of cells within the cryptopatch show little motility.

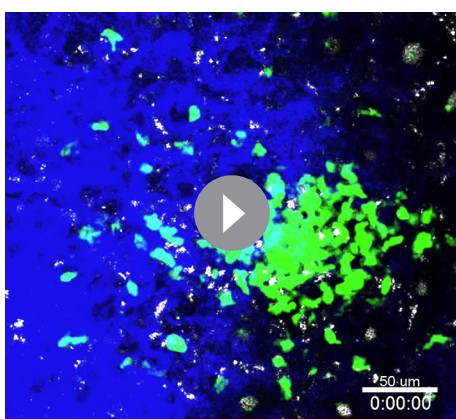

**Video 3.** ILCs within the top 15 μm of the cryptopatch are motile under steady-state conditions.

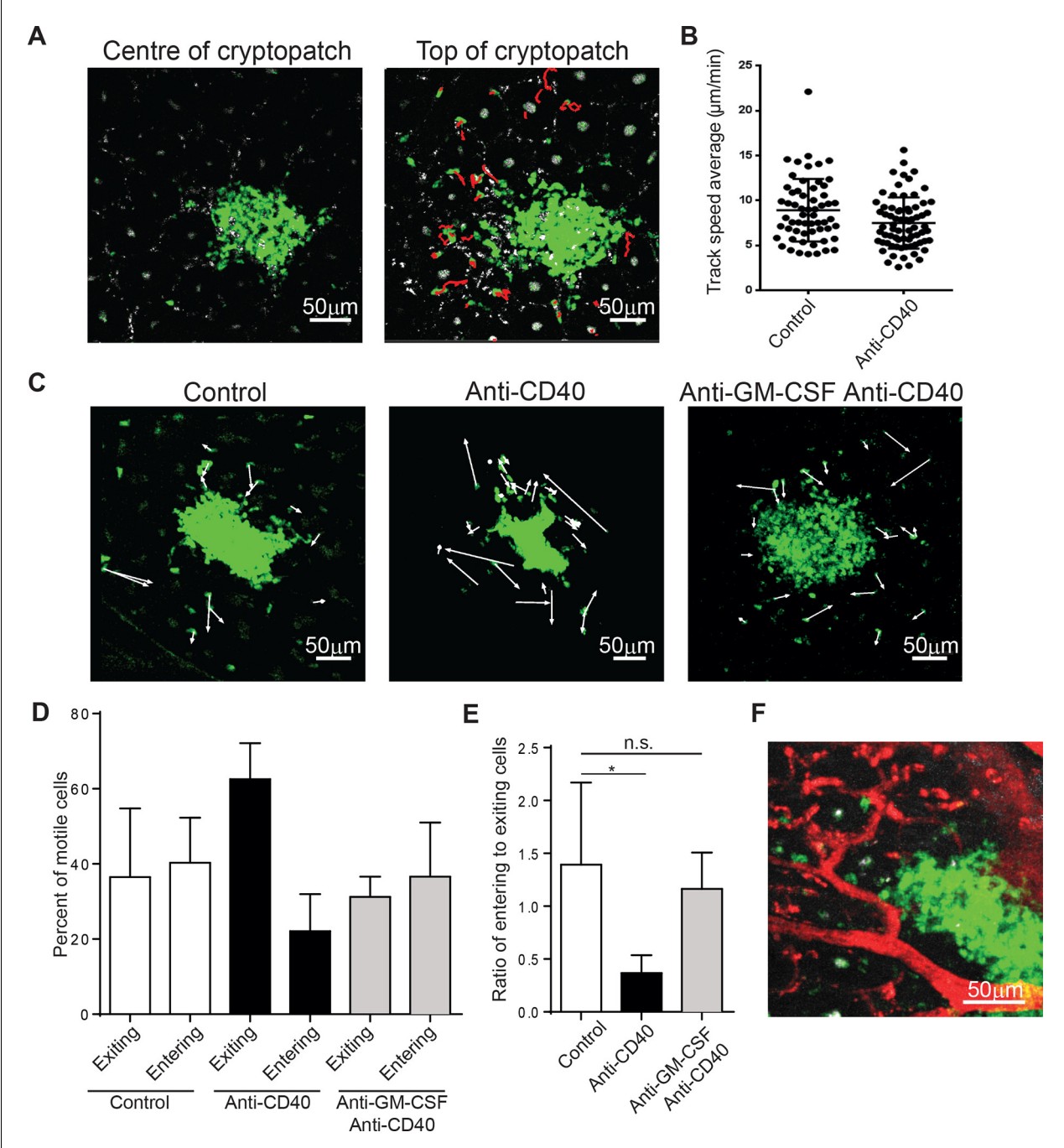

**Figure 5.** ILC3s are dynamic and mobilize after anti-CD40 treatment. (**A**) Tracks of motile ILCs in the centre (left) and superficial (right) 15 µm of a representative cryptopatch. (**B**) Track speed average of ILCs combined from 5 independent experiments before and 4-–6 hr after anti-CD40 treatment. Mean steady-state 8.9 µm/min anti-CD40 7.5 mm/min, *p*=0.01. (**C**) Displacement vectors of ILCs moving into and out of representative cryptopatches from steady state (left), 4–6 hr anti-CD40 treated mice (middle), and 4–6 hr anti-CD40 treated mice given anti-GM-CSF 24 hr before imaging (right). (**D**) Quantification of displacement of motile ILCs from control, anti-CD40 treated, and anti-GM-CSF/anti-CD40 treated mice. (**E**) Ratio of entering to exiting ILCs from control, anti-CD40, and anti-GM-CSF/anti-CD40 treated mice. Motile ILCs are defined as cells with tracks lasting more than 5 min, within 75 µm of a cryptopatch, and displacing more than 14 µm (approximately two cell lengths). *, *p*<0.05 One-way ANOVA with Tukey's post test. Data are combined (n=3–5) from at least 3 independent experiments. (**F**) Representative image of blood vessels (red) and ILCs (green) showing ILCs present adjacent to but outside blood vessels.

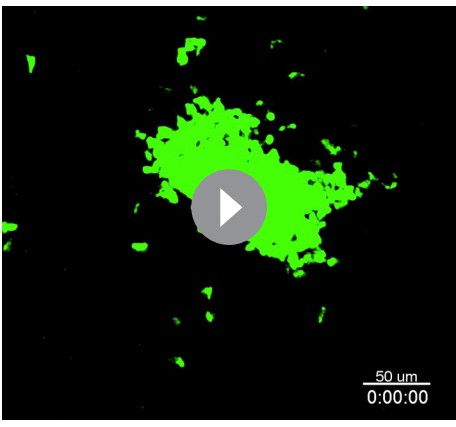

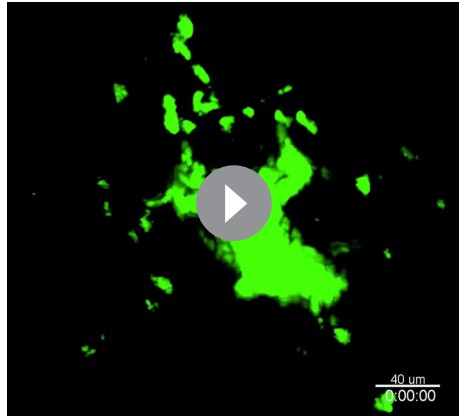

**Video 4.** Timelapse of steady-state ILCs showing tracks of motile cells.

**Video 5.** Timelapse of ILCs 6 hr after anti-CD40 treatment showing tracks of motile cells.

initiation of the immune response in the cryptopatch, followed by accumulation of immune cells within anatomically distinct villus inflammatory foci.

To understand what impact the early mobilization of ILCs has on the tissue, we looked at several chemokines that could be influencing the development of inflammatory foci. 24 hr after anti-CD40 treatment the chemokines *Ccl19* and *Ccl2* were increased in the tissue. The receptor for *Ccl19*, *Ccr7*, was also increased (*Figure 6A*). Blocking GM-CSF prevented the increase of *Ccr7* and *Ccl2*, supporting the idea that there is cross-talk between the GM-CSF and migration axes. Staining for CCR7 and CCR2 on lineage+ and ILC subsets suggested that it is the lineage positive cells that are capable of responding to the increased tissue chemokines (*Figure 6B*). Histological analysis of the proximal colon 7 days after anti-CD40 treatment showed immune foci at the tips of villi (*Figure 6C*). Immunofluorescence staining for CD11c and RORγt showed colocalization of ILCs with the CD11c positive immune infiltrate. The latter may reflect the inflammatory monocytes identified by flow cytometry from colon digests (*Figure 6D* and *1E*). Quantification of cryptopatch cellularity from H&E after anti-CD40 treatment showed a decrease in cellularity after 24 hr (*Figure 6E*). The net egress of cells from cryptopatches over a short imaging window (*Figure 5D*) corresponds to the loss of cells from the tissue over 24 hr. This decrease is followed by an increase in inflammatory infiltrates starting one day later (*Figure 6F*). Blocking GM-CSF resulted in maintenance of cryptopatch structure as would be predicted from the imaging data (*Figure 6G*). Taken together, these data indicate that ILC mobilization from cryptopatches is associated with a decrease in cryptopatch cellularity, increase in tissue chemokines, recruitment of inflammatory cells, and development of inflammatory foci associated with active disease.

## Discussion

The results of this study reveal a new IL-23-driven GM-CSF axis that promotes ILC3-mediated acute colitis. Our data indicate a spatiotemporal model for innate colitis that involves cellular activation, GM-CSF dependent mobilization of IL-23R+ ILCs from cryptopatches, and subsequent perpetuation of the response through separate ILC-containing inflammatory foci and GM-CSF dependent accumulation of inflammatory monocytes.

Although IL-23 dependent, we found that anti-CD40 induced colitis was IL-17A and IL-17F

**Video 6.** Timelapse of ILCs pretreated with anti-GM-CSF 24 hr and anti-CD40 6 hr before imaging.

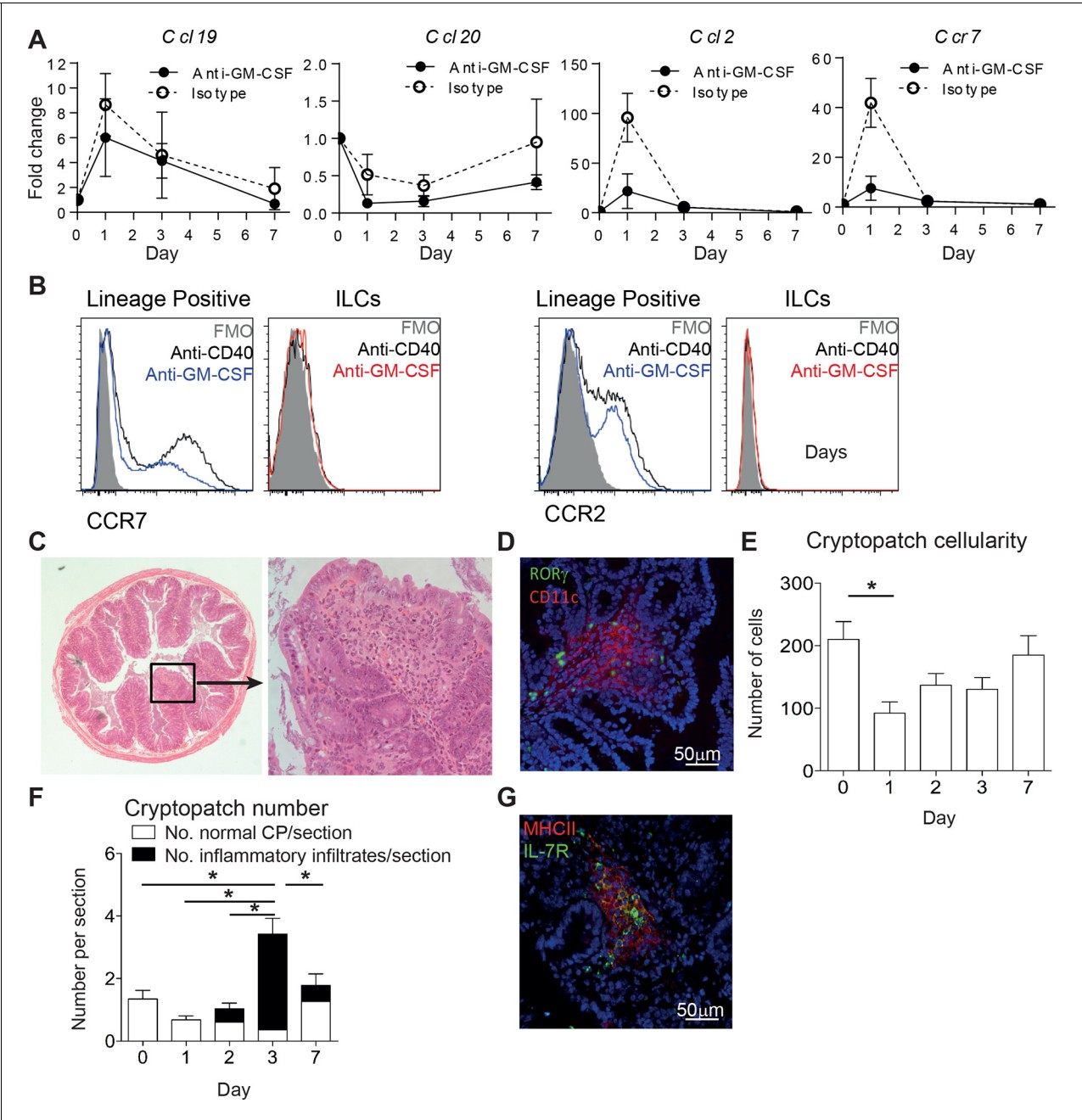

**Figure 6.** ILC movement precedes tissue reorganization. (**A**) mRNA expression of *Ccl19, Ccl20, and Ccl2* with the receptor *Ccr7* in proximal colonic lamina propria at various time points following anti-CD40 and isotype or anti-GM-CSF injection at days -1, +1, and +3. Results are shown as fold change in target gene relative to *hprt* compared with day 0 uninjected mice. (**B**) Representative flow cytometry staining of CCR7 and CCR2 in single live lineage[+] cells or ILCs from anti-CD40 or anti-GM-CSF and anti-CD40 treated mice. (**C**) Representative H&E staining of inflammatory foci (IF) in transverse proximal colon sections in B6*Rag1*[-/-] mice at day 7 following anti-CD40 injection at 2.5x and 20x magnification. (**D**) Representative immunofluorescence of IF in the proximal colon of anti-CD40 treated mice showing CD11c (red), RORγt (green) and DAPI (blue). (**E**) Number of cells in cryptopatches following anti-CD40 injection as assessed by H&E stained transverse colon sections. (**F**) Mean number of normal cryptopatches (CP) and areas containing inflammatory infiltrate in H&E stained transverse colon sections. (**G**) Representative immunofluorescence of a cryptopatch from an anti-GM-CSF and anti-CD40 treated mouse at day 3 showing MHCII (red), IL-7Rα (green), and DAPI (blue). Data are shown as means and SEM. Results are representative of 3–6 independent experiments with n=2–5 mice per experiment. *, *p*<0.05 One-way ANOVA with Bonferroni's post test. Data are shown as means and SEM. Results are pooled from 2 independent experiments with n=4–6 mice per group per experiment.

independent in keeping with earlier reports (*Figure 1* and *Eken et al., 2014*). Two recent reports indicated a functional role for IL-22, possibly through neutrophil recruitment (*Eken et al., 2014*; *Song et al., 2015*). However our data show no impact of IL-22 blockade or neutrophil depletion (data not shown) on the incidence or severity of disease in this model. By contrast, double blockade of IL-17A and IL-22 reduced colitis severity, indicating that these cytokines may play a redundant role in driving intestinal inflammation possibly through shared effects on intestinal epithelial cells (*Liang et al., 2006*). The pathogenic role of IL-22 in CD40-triggered colitis may depend on the presence of microbial factors that also influence IL-17A production in the intestine. Consistent with this, IL-22 alone was required for the maintenance of bacteria driven innate colitis-associated cancer (*Kirchberger et al., 2013*). Together these results suggest that context-dependent effector functions mediate the colitogenic potential of IL-23.

Recent studies have focussed on the role of GM-CSF as an important mediator of IL-23 driven tissue inflammation (*Codarri et al., 2011*; *El-Behi et al., 2011*; *Griseri et al., 2012*). We found that GM-CSF played a non-redundant role in both systemic and intestinal inflammation in the anti-CD40 model and was required for bacteria-induced innate colitis. The role of GM-CSF in anti-CD40 mediated systemic disease has been recently reported (*Song et al., 2015*) but effects on colitis were not assessed. Our results indicate that IL-23-responsive ILCs can act as the main source of GM-CSF in an innate model. Importantly we also find that ILCs are potent sources of GM-CSF in humans, consistent with a previous report describing cytokine-driven GM-CSF production from an NKp44⁻ ILC population (*Glatzer et al., 2013*). Our study goes further to show that GM-CSF is produced by ILC3s in the blood and colon of patients. The percent of GM-CSF$^+$ ILCs is larger than the percent of GM-CSF$^+$ T cells. This GM-CSF$^+$ ILC3 population is increased in the blood and colonic tissue of patients with IBD, indicating commonalities between mouse models and human disease pathogenesis.

IL-23 and GM-CSF may act in an autocrine loop early in the early inflammatory response as we found that anti-CD40-induced early colonic IL-23 expression was GM-CSF dependent and that in turn, IL-23 was required for sustained GM-CSF production by ILCs. A similar positive feedback loop has been described for T cell derived GM-CSF in CNS inflammation (*El-Behi et al., 2011*). Such a feedback loop would promote more GM-CSF production from ILCs as IL-23 production increases in disease. The role of GM-CSF may be to recruit myeloid cells to sites of inflammation, and impact their ability to coordinate inflammatory responses, as has recently been shown in EAE (*Croxford et al., 2015*). Indeed, we found blocking GM-CSF prevented the accumulation of Ly6C$^+$ inflammatory monocytes following anti-CD40 stimulation, in line with recent evidence for a key role for these cells in driving pathogenesis during anti-CD40 induced colitis (*Song et al., 2015*). However, GM-CSF expression also has wider consequences since it affects the behaviour of ILCs within the cryptopatch.

Functional analyses of ILC populations have mainly been restricted to cytokine production, and recent evidence that they do not recirculate suggested a primarily sedentary role for these cells (*Gasteiger et al., 2015*). However, that study did not assess ILC movement within tissues as a mechanism through which ILCs may modulate the immune and inflammatory response. To investigate this, we utilized *intra vital* microscopy to image ILC movement within cryptopatches under homeostatic conditions and in anti-CD40 induced intestinal inflammation. Cryptopatches may provide a platform for the rapid amplification of the immune response and may provide a potential mechanism by which a very small number of ILCs could initiate an immune cascade that culminates in colonic inflammation. Because of the large overlap of cell surface molecule and transcription factor expression between ILCs and T cells, imaging *Rag*$^{-/-}$ mice provided the cleanest system to study ILC motility. The anti-CD40 model provided temporal synchronization, which is a feature unique to this model of colitis. To our surprise, a subset of ILCs associated with the cryptopatch was migratory under steady-state conditions. After anti-CD40 treatment, ILC motility shifted toward egress from the cryptopatch. In agreement with data that suggests they do not recirculate through the blood (*Gasteiger et al., 2015*), ILCs were not observed migrating into the peripheral blood but appeared to crawl directly through the tissue, possibly following a local chemokine gradient. Future studies will be required to determine what factors govern entrance and exit from lymphoid structures within the tissue, which appears very different from the lymph node. Further understanding of this response may identify pathways that can be targeted to break the proinflammatory cycle within the tissue. As tools become available, studying ILC behaviour in lymphoreplete hosts will further expand these

findings and allow increased interrogation of cellular interactions between ILCs and other immune cells as well as spatio-temporal control points in the inflammatory response.

Consistent with chemokine-induced migration of ILCs, we found an early peak in mRNA expression of the chemokines *Ccl2, Ccl19* and its receptor *Ccr7* in the tissue, suggesting an early role for CCR7 and CCR2. Despite data showing that migration of LTi ILC3s from the gut to the mesenteric lymph nodes is dependent on CCR7 expression (*Mackley et al., 2015*), expression of CCR7 on ILCs was not assessed in that study and we were unable to identify CCR7 or CCR2 on intestinal ILC3s in anti-CD40 induced colitis. While this argues against CCR7 driven chemotaxis of the ILCs, it does not preclude an organizational function for ILCs in the tissue driven by other mediators. Indeed, our data indicate a role for GM-CSF in the net egress of ILCs from cryptopatches in anti-CD40 induced colitis. Similarly, increased amounts of *ccl2* and *ccr7* mRNA in the colon were also GM-CSF-dependent, as was accumulation of inflammatory monocytes which also express CCR7 and CCR2 (*Tsou et al., 2007*; *Förster et al., 2008*). Together, the data are compatible with a model in which GM-CSF-dependent activation of ILCs leads to exit from the cryptopatch to the villus tip. This coincides with the recruitment of CCR7 and CCR2 expressing inflammatory monocytes through a CCL19 and CCL2 gradient, culminating in tissue damage. Sustained production of GM-CSF by ILCs in inflammatory foci could recruit inflammatory progenitor cells and contribute to differentiation of myeloid effector cells. GM-CSF has been shown to control the differentiation of inflammatory monocytes (*Lenzo et al., 2012*), and this population was markedly reduced following anti-GM-CSF treatment. GM-CSF may also impact on this pathway through altered haematopoiesis and accumulation of granulocyte macrophage progenitors as described in T cell dependent colitis (*Griseri et al., 2012*). Given the multiple activities of GM-CSF on myeloid cell development and function, further studies will be required to delineate the key downstream pathways.

IL-23 has been suggested as a therapeutic target in IBD (*Bowman et al., 2006*; *Uhlig et al., 2006*). However, our studies indicate that neutralisation of GM-CSF is as efficacious as blockade of IL-23 in mouse models. In addition, increases in GM-CSF secreting ILCs are a feature of IBD. However, targeting GM-CSF may not be straightforward in IBD as there are clear host protective functions of GM-CSF in the intestine (*Sainathan et al., 2008*; *Bernasconi et al., 2010*; *Hirata et al., 2010*). Indeed neutralising anti-GM-CSF autoantibodies are increased in IBD, particularly in patients with ileal and stricturing disease (*Han et al., 2009*). Furthermore, administration of recombinant GM-CSF (sargramostim) has been used to treat Crohn's disease (*Korzenik et al., 2005*) although further trials did not show any substantial benefit of GM-CSF treatment (*Roth et al., 2012*). Despite these issues, in light of recent developments testing anti-GM-CSF in clinical trials for the treatment of other inflammatory conditions such as multiple sclerosis (*Constantinescu et al., 2015*) it will be interesting to see if subgroups of patients with hyperinflammatory innate immune activation-driven IBD could benefit from GM-CSF blockade.

## Materials and methods

### Human study subjects

IBD and colorectal cancer (CRC) patients were recruited through the Oxford IBD cohort study, and samples were obtained from Oxford GI Biobank in collaboration with Oxford Radcliffe Biobank. Colon specimens were collected from patients undergoing surgery for severe disease. Macroscopically healthy sections from CRC patients were used as controls. For gene expression data, intestinal mucosal pinch biopsies were taken from patients undergoing routine endoscopy. From 20 patients with evidence of active intestinal inflammation, matched biopsies were collected from both inflamed tissue and regions of the intestine with no apparent inflammation. IBD blood samples were collected from patients attending the outpatient clinic and in some cases from patients undergoing surgery. Control blood samples were obtained from healthy donors. Ethical approval was obtained from the Oxfordshire Research Ethics Committee (Reference numbers: 11/YH/0020 and 09/H0606/5) and informed written consent was given by all study patients.

A previously published study with a publically available dataset (Arijs I., Van Lommel L., Van Steen K., De Hertogh G., Geboes K., Schuit F. and Rutgeerts P. 2009. Mucosal expression profiling in patients with inflammatory bowel disease before and after first infliximab treatment. Accession number GSE16879, Gene Expression Omnibus repository) was used to confirm mRNA observations of

the Oxford cohort, using colonic mucosal biopses obtained at endoscopy from healthy controls or patients refractory to corticosteroids and/or immunosuppression. Gene expression data were generated using the Affymetrix Human Genome U133 Plus 2.0 array as described in (*Arijs et al., 2009*).

## Mouse strains

All mice were bred and maintained under specific pathogen-free conditions in accredited animal facilities. C57BL/6 $Rag1^{-/-}$, 129SvEvS6.$Rag2^{-/-}$ and C57BL/6 $Rag1^{-/-}Il23r^{gfp/+}$ (from Daniel Cua, Merck Research Laboratories, Palo Alto, USA) strains used in this study were bred and maintained at the University of Oxford. C57BL/6 $Rag1^{-/-}Il15ra^{-/-}$ and C57BL/6 $Rag1^{-/-}$ were bred and maintained at the National Institute for Medical Research. Experiments were conducted in accordance with local animal care committees (UK Scientific Procedures Act of 1986). Mice were routinely screened for the absence of pathogens, and were kept in individually ventilated cages with environmental enrichment.

## Isolation of murine lamina propria leukocytes

For cLP isolation, colon tissue was cut into 1cm pieces, and epithelial cells were removed by incubation (2x) in RPMI containing 5% heat-inactivated FCS and 5mM EDTA. Remaining tissue was incubated in RPMI containing 5% FCS, 15mM HEPES and 300U/ml of Collagenase VIII (Sigma-Aldrich, St Louis, MO) to digest the remaining tissue. Cell populations were purified by 75%:40%:30% Percoll (GE Healthcare, Little Chalfont, U.K.) gradient centrifugation (600 x $g$, 20 min). Lymphocytes were isolated from the 40%/30% interface.

## Isolation of human peripheral blood mononuclear cells (PBMCs)

Peripheral blood was diluted in an equal volume of PBS and layered onto Ficoll-Hypaque (GE-Healthcare). Following centrifugation, PBMCs were collected from the Ficoll-plasma interface and washed 3x in PBS prior to culture or flow cytometry.

## Isolation of human lamina propria mononuclear cells (LPMCs)

Specimens were incubated in 1 mM DTT solution, followed by three washes in 0.75 mM EDTA solution. Tissues were then digested overnight in 0.1 mg/ml collagenase A solution (Roche Diagnostics, Basel, Switzerland). Cell populations were purified by 100%:60%:40%:30% Percoll (GE Healthcare) gradient centrifugation (900 x $g$, 30 min). Lymphocytes were isolated from the 60%/40% interface.

## Quantification of mRNA levels by real-time PCR

For human biopsy samples, tissue was immediately placed in RNA*later* solution (ThermoFisher Scientific, Waltham, MA) and stored at -20°C until processed. Human and murine cLP cells were lysed in RLT buffer (Qiagen, Hilden, Germany) containing ß-mercaptoethanol. RNA was isolated using the RNEasy kit (Qiagen) including a DNAse I digestion. Content and purity of RNA was measured using a Nanodrop spectrophotometer (Thermo Fisher Scientific). cDNA was synthesised using the Superscript III reverse transcription kit (Life Technologies, Paisley, U.K.) or the High Capacity cDNA kit (ThermoFisher Scientific, Waltham, MA). Quantitative real-time PCR for the candidate genes was performed using the Taqman system. cDNA samples were analysed in triplicate (technical replicates) using the CFX96 detection system (Bio-Rad Laboratories, Hercules, CA) or the Viia7 system (Applied Biosystems, Foster City, CA). Values were normalised to *Hprt* expression, and analysed using the d-Ct method. Taqman Gene Expression Assays (Life Technologies) for mouse *Ccl19, Ccl20 Ccr7, Ccl2, Il23r* and *Csf2*, and human *RPLPO* and *CSF2* were used.

## Induction of colitis

To induce acute innate colitis, 25-–50 mg of αCD40 IgG2a mAb (clone FGK45, BioXCell, West Lebanon, NH) was administered via i.p. injection to mice on a C57BL/6 $Rag^{-/-}$ background. Unless otherwise indicated, mice were sacrificed 7 days post αCD40 injection, and weight loss was monitored throughout the course of the experiment.

To induce colitis using *Helicobacter hepaticus, Hh* NCI-Frederick isolate 1A (strain 51449) was grown as previously described (*Maloy et al., 2003*). Mice on a 129SvEv $Rag2^{-/-}$ background were

fed 3x on consecutive days with *Hh* 1A (1.0x10$^8$ CFU) by oral gavage. Animals were analysed at week 6 following infection.

## In vivo antibody treatment

To block specific cytokine or cellular activity in vivo, aCD40-treated mice received 0.375 mg IL-17A blocking mAb (UCB Celltech, Slough, U.K.), 0.15 mg IL-22 blocking mAb (Genentech, San Francisco, CA, clone 8E11), 0.8 mg IL-23R blocking mAb (D. Cua, Merck, Kenilworth, NJ), 0.125 mg GM-CSF blocking mAb (CSL Ltd, Parkville, Australia), 100 mg IL-17F blocking antibody (eBioscience, San Diego, CA) or isotype controls (GP120 10E7.1D2, Genentech or 22E9.11, CSL Ltd). Treatment started on day -1 and antibodies were injected i.p. at day -1 and day 3, except blocking mAb to IL-22 which was administered at days -1, 1, 3 and 5. Blocking mAb to GM-CSF during *Helicobacter hepaticus* infection was administered i.p. 2x per week starting at day 0.

## Mouse antibodies

The following antibodies were purchased from eBioscience: fixable viability dye, GM-CSF-PE (MP1-22e9), CD127-FITC/PE/PECy5 (A7R34), Thy1.2-PECy7 (53.2.1), T-bet-eFluor660 (eBio4B10), RORγt-PE (AFKJS-9), B220-PerCP-Cy5.5 (RA3-6B2), CD11b-PerCP-Cy5.5 (M1/70), CD11c-PerCP-Cy5.5/APC (N418), Gr-1-PerCP-Cy5.5 (RB6-8C5), CD64-PE (X54-5/7.1), F4/80-PerCP-Cy5.5 (BM8), MHCII-Alexa-Fluor700 (M5/114.15.2), CCR7-PE (4B12), perforin-APC (eBioOMAK-D) and cKit-Alexa-Fluor700 (ACK2). The following antibodies were purchased from Biolegend (San Diego, CA): NKp46-FITC (29A1.4), CD49b-PerCP-Cy5.5 (DX5), Ly6C-PECy7 (HK1.4), Ly6G-BV570 (1A8), CCR6-BV421 (29-2L17), CD4-BV605 (RM4-5), CCR2-AF647 (SA203G11) and NK1.1-BV650 (PK136). The following antibodies were purchased from BD Horizon (Franklin Lakes, NJ): NKp46-V450 (29A1.4), CD11b-V500 (M1/70), CD45-V500/BV650 (30-F11), SiglecF-BV421 (E5-2440) and NKG2D-PE CF594 (CX5). Lineage staining was performed using PerCP-Cy5.5-conjugated antibodies against B220, CD11b, CD11c, Gr-1, F4/80 and CD49b.

## Human antibodies

The following antibodies were purchased from eBioscience: CD1a-FITC (HIH9), CD3-BV650 (OKT3), CD8-FITC (RPA-T8), CD19-FITC (HIB19), IL-17A-PE (eBio64DEC17), IL-22-PECy7 (22URT1). The following antibodies were purchased from Biolegend: CD3-FITC (UCHT1), CD4-FITC (OKT4), CD11c-FITC (B-LY63.9), CD14-FITC (MSE2), CD16-FITC (3G8), CD34-FITC (581), CD123-FITC (6H6), TCRαβ-FITC (IP26), TCRγδ-FITC (B1), CD45-AlexaFluor700 (HI30), CD56-PerCP-Cy5.5/BV510 (HCD56), CD127-BV421 (A019D5), GM-CSF-PerCP-Cy5.5 (BVD2-21C11), IL-5-APC (TRFK5) and IFNγ-BV605 (4S.B3). Lineage staining was performed using FITC-conjugated antibodies against CD1a, CD8, CD19, CD3, CD4, CD14, CD16, CD34, CD123, TCRαβ and TCRγδ.

## Flow cytometry

Single cell suspensions were prepared from mouse spleen or colonic LP or human LPMC or PBMC cells as described above. For intracellular cytokine staining mouse or human were directly cultivated for 3 or 4 hr in the presence of phorbol 12-myristate 13-acetate (PMA, Sigma-Aldrich) (100ng/ml), ionomycin (Sigma-Aldrich) (1µg/ml), monensin (BD Biosciences) and brefeldin A (eBioscience). Where indicated, mouse and human cLP cells were stimulated overnight with 10 ng/ml of recombinant IL-12 (Peprotech, Rocky Hill, NJ) or IL-23 (mouse - R&D Systems (Minneapolis, MN), human - Peprotech), and then for the last 4 hr PMA, ionomycin and brefeldin A were added.

Cells were incubated with CD16/32 (eBioscience) to block Fcγ receptors, prior to incubation with fixable viability dye (eFluor780, eBioscience), and mAb conjugates in 100 µl PBS containing 2% BSA and 1 mM EDTA (PBS-E) for 30 min. Cells were subsequently washed twice in PBS-E. For cytokine staining, cells were stained with antibodies for 1 hr using the Cytofix/Cytoperm kit (BD Biosciences). For transcription factor staining, cells were stained with transcription factor antibodies for 1 hr using the Foxp3 staining kit (eBioscience). Cells were washed twice in PBS-E. All labelled cells were then acquired on a LSRII SORP (BD Biosciences) and analysed using Flowjo software (Tree star, Ashland, OR).

## Histology and pathology score

To assess the severity of colitis, samples of the proximal colon were taken and immediately fixed in buffered 10% formalin (3.6% w/v formaldehyde). 4–5 µm paraffin-embedded sections were cut and stained with haematoxylin and eosin (H&E). Inflammation was assessed using previously published criteria (*Izcue et al., 2008*). Each sample was graded semiquantitatively from 0 to 4 in each of the 4 following features: epithelial hyperplasia and goblet cell depletion, cellular infiltration of the lamina propria, percentage of the section affected, and markers of severe inflammation (submucosal inflammation, crypt abscesses). Three separate colon sections for each sample were examined. Scores for each criterion were added to give an overall score for each sample of 0–12. Samples were scored in a blinded fashion by two individuals.

## Intravital microscopy

Gut imaging was performed as described in McDole *et al*. (*McDole et al., 2012*). Briefly, mice were anesthetized with ketamine/xylazine, the gut was exposed and immobilized with an imaging window, and imaging proceeded using isoflurane to maintain anesthesia. Images were collected using a 20x water dipping lens and the spectral detector of a Zeiss 780 upright microscope (Carl Zeiss, Oberkochen, Germany). Images were linearly unmixed using the Zen software (Carl Zeiss) to separate autofluorescence, collagen, EGFP, and Texas red-dextran based on single color controls.

## Blood vessel labelling

Mice were anesthetized for imaging and injected intravenously with 50 mg of 70 kDa texas-red labelled dextran (Life Technologies). Imaging proceeded immediately as described above.

## Image analysis

Intravital microscopy was analyzed using Imaris 8 (Bitplane, Belfast, U.K.) after linear unmixing using Zen (Carl Zeiss). Images were drift corrected based on mucus or collagen signal. ILCs were tracked using autoregressive motion algorithm of the spot-tracking module. Motile ILCs were gated based on distance less than 75 µm from a cryptopatch, track duration greater than 5 min, and displacement greater than 14 µm (approximately two cell lengths). Images were smoothed using a Gaussian filter for display.

## Immunofluorescence

For immunofluorescence, samples of the proximal colon were taken and immediately embedded in OCT compound (Tissue-Tek, Sakura, Alphen aan den Rijn, The Netherlands) and frozen in a bath of isopentane on dry ice prior to storage at -80°C. 6 µm frozen sections were cut and collected on frosted glass slides. For staining, slides were fixed in 2% formalin. For RORγt staining, slides were subsequently also treated with Fix-Perm buffer (eBioscience). Endogenous peroxidase activity was blocked with 1% $H_2O_2$ (Sigma-Aldrich) and 2% sodium azide and non-specific binding was blocked with 10% donkey serum. Sections were incubated with the following antibodies CD4 (RM4-5, BD Biosciences) IL-7Rα (A7R34, eBioscience) in 10% donkey serum or RORγt (AFKJS-9, eBioscience) in Perm buffer (eBioscience). Sections were then incubated with donkey anti-rat HRP secondary antibody (Jackson ImmunoResearch Laboratories, West Grove, PA). Tyramide signal amplification was the performed (Perkin Elmer, Waltham, MA). Then sections were stained with CD11c (N418, eBioscience) in 10% goat serum followed by goat anti-hamster Cy3, or with MHCII-FITC or AF647 (both M5/114.15.2, Biolegend).

Sections were mounted with Vectashield containing DAPI. Images were collected using a 710 microscope (Carl Zeiss), and analysed using ImageJ open access software.

The colons from C57BL/6 *Rag1*[-/-] *Il23r*[gfp/+] mice were freshly isolated and maintained in room temperature PBS before imaging. Opened sections of colon were immobilized using Vetbond tissue adhesive (3M, Saint Paul, MN) in dishes containing PBS. CPs were imaged from the muscularis side of the colon using a 20x water dipping objective and spectral detector on a Zeiss 780 upright multiphoton microscope. Images were analysed using the linear unmixing module of Zen (Carl Zeiss) and Imaris (Bitplane).

## Statistical analysis

Statistical analysis and graphical representations were performed using Prism 5.0a software (Graph-Pad, La Jolla, CA). For comparison of qPCR results and histopathological analysis, the nonparametric Mann-Whitney U test was used, and for multiple samples, one-way ANOVA with Bonferroni's post-test was used unless otherwise stated. Technical replicates are from the same sample, analysed on the same day. Biological replicates are from independent experiments.

## Sample size estimation

Sample size was computed using an estimate of colitis score between 6–8, to detect a score reduction by 50% in a treated group with a standard deviation of 30%, false positive of 0.05, and power of 0.80. Sample size was therefore calculated to be between 6–7. Where smaller changes were expected, sample sizes were increased accordingly.

## Acknowledgements

We would like to thank R Stillion and the Kennedy Institute of Rheumatology histology department for histology, and the staff of our animal houses for their assistance; S Kirchberger, M Hühn, A Hegazy, A Chauveau, and T Arnon for technical advice and discussion; C Lagerholm and the Wolfson Imaging Centre for microscopy assistance; B Owens and C Arancibia for critically reading the manuscript; W Ouyang (Genentech) for the blocking anti-IL-22 antibody and isotype controls; UCB Celltech for providing the blocking anti-IL-17 antibody; D Cua (Merck) for providing the blocking anti-IL-23R antibody. We acknowledge the contribution to this study made by the Oxford Centre for Histopathology Research and the Oxford Radcliffe Biobank, which are supported by the NIHR Oxford Biomedical Research Centre. We would like to thank all patients and investigators who contributed to the Oxford IBD cohort study and the Oxford GI Biobank.

C Pearson, E Thornton, T Griseri and F Powrie were supported by the Wellcome Trust, A-L Schaupp by an NDM studentship. N West was supported by the Cancer Research Institute. B Seddon and S Tung were supported by the Medical Research Council. N Huskens performed this work as part of an MSc studentship. H Uhlig was supported by the Crohn's and Colitis Foundation of America (CCFA), The Leona M. and Harry B. Helmsley Charitable Trust, ForCrohn's and Crohn's and Colitis UK . FP and HU have no conflict of financial interest related to the article but non-related previous or current project collaborations with industry that included Eli Lilly, UCB Pharma, Novartis, Merck and Pfizer. B McKenzie was employed by CSL Ltd and is currently employed by Genentech, Inc.

## Additional information

### Competing interests

FP: Reviewing editor, *eLife*. The other authors declare that no competing interests exist.

### Funding

| Funder | Grant reference number | Author |
| --- | --- | --- |
| Wellcome Trust | Investigator Award, 095688/Z/11/Z | Claire Pearson Emily E Thornton Nicky Huskens Thibault Griseri |
| University Of Oxford | NDM Studentship | Anna-Lena Schaupp |
| Cancer Research Institute | | Nathaniel West |
| Medical Research Council | | Sim Tung Benedict P Seddon |
| Crohn's and Colitis Foundation of America | | Holm H Uhlig Fiona Powrie |
| Crohn's and Colitis UK | | Holm H Uhlig |

| Leona M. and Harry B. Helmsley Charitable Trust | Holm H Uhlig |
| --- | --- |
| ForCrohns | Holm H Uhlig |

The funders had no role in study design, data collection and interpretation, or the decision to submit the work for publication.

## Author contributions

CP, EET, Conception and design, Acquisition of data, Analysis and interpretation of data, Drafting or revising the article; BMcK, Conception and design, Contributed unpublished essential data or reagents; ALS, NH, Acquisition of data, Analysis and interpretation of data; TG, Conception and design, Acquisition of data, Analysis and interpretation of data; NW, Collected human colon samples, Performed qPCR analysis and analysed the human cohort data, Acquisition of data, Analysis and interpretation of data; ST, Designed and performed the IL-15RecKO colitis experiments, Conception and design, Acquisition of data; BPS, Conceived the IL-15RecKO experiments and edited the manuscript., Conception and design, Drafting or revising the article; HHU, Conception and design, Drafting or revising the article; FP, Conception and design, Analysis and interpretation of data, Drafting or revising the article

## Ethics

Human subjects: IBD and colorectal cancer patients were recruited through the Oxford IBD cohort study, and samples were obtained from the Oxford GI Biobank in collaboration with Oxford Radcliffe Biobank. Ethical approval was obtained from the Oxfordshire Research Ethics Committee (reference numbers 11/YH/0020 and 09/H0606/5) and informed written consent was given by all study patients. Animal experimentation: This study was performed in strict accordance with the EU Animal Use Directive (Directive 2010/63/EU). The experiments were performed with approval from the University of Oxford Local Ethical Review Panel and under a Home Office project licence. All mice were bred and maintained under specific pathogen-free conditions in accredited animal facilities at the University of Oxford.

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
