## [Decision Letter]

Thank you for submitting your work entitled "Pathogenic ILC3 GM-CSF production and mobilisation orchestrate acute intestinal inflammation" for peer review at *eLife*. Your submission has been favorably evaluated by K VijayRaghavan (Senior Editor) and three reviewers, one of whom is a member of our Board of Reviewing Editors.

The reviewers have discussed the reviews with one another and the Reviewing editor has drafted this decision to help you prepare a revised submission.

This manuscript further elaborates the role ILCs play in the pathology of colitis induced by agonistic anti-CD40 antibody by showing that group 3 ILC-driven colitis in this model depends on GM-CSF. It also shows that ILCs traffic through gut cryptopatches under homeostatic conditions and that their egress is enhanced by anti-CD40 antibody, associated with increased tissue chemokine levels and development of inflammatory foci. However, before publication, the following issues would be important to address.

1) While the manuscript purports to exclude roles for IL-17A and IL-22, combined neutralisation of these two cytokines has not been done, nor has a role for IL-17F been tested. Also, the data indicate that GM-CSF neutralisation may not entirely abrogate disease, and triple-neutralisation studies would be useful to examine if there are some distinct non-overlapping quantitative roles for these various cytokines.

2) The determination of GM-CSF-producing cells is not precise. There is a significant GMCSF production by cells that weakly express lineage markers. CD127+ lineage- cells appear to contain some lineage+ cells. Have the lineage-dim cells been gated out when analyzing the ILC subsets producing GM-CSF? There is no negative control shown for IL7R staining (Figure 2), making the interpretation difficult to judge, since CD127-low cells could represent non-NK ILC1 cells that also express NK1.1 and NKp46 (Fuchs et al. 2013) and Eomes or perforin have not been used to exclude this possibility. In any case, these supposedly IL7Ra- 'NK' cells produce GM-CSF, raising the question of their contribution to disease.

3) For mechanistic insights into the anti-CD40-mediated modulation of the ability of ILC3s to produce GM-CSF, it is useful to test if the levels of Rorc or IL-23R have been modulated by anti-CD40 treatment.

4) Since a role for GM-CSF in inflammatory situations is not by itself novel (although not demonstrated for this particular situation), it is appropriate to expect some data attempting to examine possible mechanisms through which only GM-CSF appears to be non-redundantly required in a disease depending on cells that make all three potentially pro-inflammatory cytokines, IL-17, IL-22 and GM-CSF. Notably, the authors show data that GM-CSF appears to be required for the induction of a putative upstream mediator (of IL-17 and IL-22 as well as GM-CSF), IL-23, making a number of complex regulatory scenarios plausible and worth examination. Also, there is prior evidence that bacterial gut inflammation may require IL-22, at least, and comparative data on the relative roles of these inflammatory cytokines between non-microbial induction of colitis as used in the manuscript and, say, *Helicobacter hepaticus*-induced colitis (a model in use by the authors) would be very helpful.

5) Figure 3 is not very informative as it stands. It is already known that human NKp44^+^ ILC3 produce GM-CSF (Glatzer et al. 2013), and the present data do not identify the subsets of total ILCs that produce GM-CSF. While more ILCs from the peripheral blood of IBD patients appear to express GM-CSF, the authors suggest in the text that this is not so for IFNγ, while the data in Figure 3 appear to show more IFNγ-producing ILCs as well. Further, there are no data regarding GM-CSF expression in healthy versus IBD-inflamed colonic tissue. Thus, these data are correlative and say little about a potential role of GM-CSF in human IBD. These ambiguities need to be addressed explicitly.

6) With respect to the data on ILC migration (Figure 6), it would be useful to clarify if the authors are referring to aggregates in both villous tips and in crypts as inflammatory foci. It would also help to know if ILC3s express CCR2 and/or CCR7.

7) The manuscript suggests that migration of innate-like lymphoid cells is essential for colitogenesis, and that one role of GM-CSF is in controlling this migration by inducing the expression of CCR7, the receptor for the chemokine CCL19. Yet, the manuscript shows no studies examining, for example, the balance of cryptopatch-egress in animals with GM-CSF blockade or with CCL19 or CCR7 blockade, or indeed examining possible modulations in the state of the villous-tip inflammatory foci by CCL19 blockade, which would help provide evidence for these putative cause-and-effect relationships between innate-like lymphocyte migration and inflammogenesis.

---

## [Author Response]

This manuscript further elaborates the role ILCs play in the pathology of colitis induced by agonistic anti-CD40 antibody by showing that group 3 ILC-driven colitis in this model depends on GM-CSF. It also shows that ILCs traffic through gut cryptopatches under homeostatic conditions and that their egress is enhanced by anti-CD40 antibody, associated with increased tissue chemokine levels and development of inflammatory foci. However, before publication, the following issues would be important to address. 1) While the manuscript purports to exclude roles for IL-17A and IL-22, combined neutralisation of these two cytokines has not been done, nor has a role for IL-17F been tested. Also, the data indicate that GM-CSF neutralisation may not entirely abrogate disease, and triple-neutralisation studies would be useful to examine if there are some distinct non-overlapping quantitative roles for these various cytokines.

We have now included IL-17A/IL-22 double blockade as well as IL-17F blockade (Figure 1—figure supplement 1). IL-17F blockade did not reduce disease. Double blockade of IL-17A and IL­22 did reduce disease severity, indicating that these cytokines may play a redundant role in driving pathogenesis. The level to which blocking IL-22 ameliorates colitis may depend on microbial factors that also influence IL-17A production in the intestine. Indeed as pointed out by the reviewer our own studies revealed a non-­redundant role for IL­22 in bacteria driven innate colitis-associated cancer (Kirchberger et al. 2013). These findings illustrate the complex contextual role of IL-22 in intestinal pathophysiology and this point has now been included in the Discussion.

2) The determination of GM-CSF-producing cells is not precise. There is a significant GMCSF production by cells that weakly express lineage markers. CD127+ lineage- cells appear to contain some lineage+ cells. Have the lineage-dim cells been gated out when analyzing the ILC subsets producing GM-CSF? There is no negative control shown for IL7R staining (Figure 2), making the interpretation difficult to judge, since CD127-low cells could represent non-NK ILC1 cells that also express NK1.1 and NKp46 (Fuchs et al. 2013) and Eomes or perforin have not been used to exclude this possibility. In any case, these supposedly IL7Ra- 'NK' cells produce GM-CSF, raising the question of their contribution to disease.

We thank the reviewers for this point about ILC gating strategy, which is critical to this study. We have updated Figure 2 to focus only on GM-CSF producing ILCs. To clarify the role of lineage dim cells, we have identified them as NK cells by surface marker and perforin expression (Figure 2—figure supplement 1). By using *Rag1^-/-^ l15ra^-/-^*we now show that lack of NK cells, while it does have an effect on weight loss, does not significantly affect colitis (Figure 2—figure supplement 1).

*3) For mechanistic insights into the anti-CD40-mediated modulation of the ability of ILC3s to produce GM-CSF, it is useful to test if the levels of Rorc or IL-23R have been modulated by anti-CD40 treatment.*

We proposed that the effect of anti-CD40 on ILCs is from myeloid cell derived IL-23, which is supported by the data in Figure 2. The analysis of RORγt and IL-23R shows a transient decrease in both molecules 3 days after anti-CD40 administration (Figure 4—figure supplement 1). To understand how quickly this effect occurs, we looked at *Il23r* expression 6 hours after anti-CD40 treatment (Figure 4—figure supplement 1). At this time point, the receptor is increased, indicating that anti-CD40 triggers an inflammatory cascade (through effects on myeloid cells) that has very early effects on ILCs, which fits nicely with the imaging data presented in Figure 5.

4) Since a role for GM-CSF in inflammatory situations is not by itself novel (although not demonstrated for this particular situation), it is appropriate to expect some data attempting to examine possible mechanisms through which only GM-CSF appears to be non-redundantly required in a disease depending on cells that make all three potentially pro-inflammatory cytokines, IL-17, IL-22 and GM-CSF. Notably, the authors show data that GM-CSF appears to be required for the induction of a putative upstream mediator (of IL-17 and IL-22 as well as GM-CSF), IL-23, making a number of complex regulatory scenarios plausible and worth examination. Also, there is prior evidence that bacterial gut inflammation may require IL-22, at least, and comparative data on the relative roles of these inflammatory cytokines between non-microbial induction of colitis as used in the manuscript and, say, Helicobacter hepaticus-induced colitis (a model in use by the authors) would be very helpful.

Our new data supports dual mechanisms for GM-CSF in initiation and perpetuation of inflammation. We now show that GM-CSF is required for the mobilization of ILCs from cryptopatches, implicating the cytokine in the pro-inflammatory initiation step of the disease. These data, coupled with a decrease in inflammatory monocytes with anti-GM-CSF blockade, newly implicate ILCs in the development of inflammatory foci within the tissue. While our manuscript was in revision, a new study appeared demonstrating a role for GM-CSF in the systemic anti-CD40 induced disease in *Rag^-/-^*mice and in accumulation of inflammatory monocytes in the intestine. There was no analysis of the functional role of GM-CSF in colitis or movement of ILCs in that study, so our data provide important new mechanistic insight into the functions of GM-CSF in the colon. However both studies together strongly support a role for GM-CSF in monocyte/macrophage accumulation in the intestine.

The inflammatory cytokines downstream of IL-23 likely play different roles in bacterial versus anti- CD40 driven colitis. Indeed anti-CD40 induced colitis occurs in germ free mice indicating the aetiopathogenesis can be microbe independent (Uhlig et al. 2006). However the presence of particular microbes in the RAG colony may alter IL-23 dependent down-stream effector mechanisms in anti-CD40 colitis. Indeed our own data supports this showing a non-redundant role for IL-23 driven IL-22 in *H. hepaticus* driven innate colitis associated cancer. This may reflect the differential involvement of epithelial derived signals in bacteria versus anti-CD40 induced colitis with IL-17 and IL-22, which act on the epithelium, having a more dominant role in the former. Importantly by contrast with the redundant role of IL-17 and IL-22 in anti-CD40 colitis, GM-CSF is required in both models and we have now included data showing anti-GM-CSF blockade ameliorates colitis in bacteria-driven innate colitis (Figure 1—figure supplement 2). A more in-depth discussion of the context dependent roles of IL-23 driven downstream cytokines, which are of relevance to therapeutic targeting in IBD, has been added to the Discussion.

*5) Figure 3 is not very informative as it stands. It is already known that human NKp44^+^ ILC3 produce GM-CSF (Glatzer et al. 2013), and the present data do not identify the subsets of total ILCs that produce GM-CSF. While more ILCs from the peripheral blood of IBD patients appear to express GM-CSF, the authors suggest in the text that this is not so for IFN*γ*, while the data in Figure 3 appear to show more IFN*γ*-producing ILCs as well. Further, there are no data regarding GM-CSF expression in healthy versus IBD-inflamed colonic tissue. Thus, these data are correlative and say little about a potential role of GM-CSF in human IBD. These ambiguities need to be addressed explicitly.*

We appreciate the reviewers’ comments regarding the human data and have added additional new data to strengthen our findings. We now include the analysis of GM-CSF by distinct subsets of blood ILCs in Figure 3. The text has been amended to clarify that inflammatory cytokines other than GMCSF are also increased in blood ILCs of IBD patients. We have included data showing that a larger proportion of ILCs express GM-CSF in the colon compared with the proportion of T cells, and the proportion of GM-CSF^+^ cells is higher in the colon than the blood (Figure 3). Data from the publicly available Leuven cohort as well as our Oxford TGU cohort show increased GM-CSF in patients with IBD with increased expression associated with increased disease severity (Figure 4). Taken together these data support a role for GM-CSF in human IBD.

6) With respect to the data on ILC migration (Figure 6), it would be useful to clarify if the authors are referring to aggregates in both villous tips and in crypts as inflammatory foci. It would also help to know if ILC3s express CCR2 and/or CCR7.

We have clarified that aggregates refer to accumulations in the villous tips. Data now included in Figure 6 shows that ILC3s in the colon do not express CCR2 or CCR7. This suggests that the increase in CCR7 and CCR2 in the tissue is a result of chemokine receptor increase within the lineage^+^ population. The question of ILC chemotaxis is therefore still an open topic, and further studies, beyond the scope of this report, will be required to elucidate what is likely a very complicated and possibly tissue-specific process.

7) The manuscript suggests that migration of innate-like lymphoid cells is essential for colitogenesis, and that one role of GM-CSF is in controlling this migration by inducing the expression of CCR7, the receptor for the chemokine CCL19. Yet, the manuscript shows no studies examining, for example, the balance of cryptopatch-egress in animals with GM-CSF blockade or with CCL19 or CCR7 blockade, or indeed examining possible modulations in the state of the villous-tip inflammatory foci by CCL19 blockade, which would help provide evidence for these putative cause-and-effect relationships between innate-like lymphocyte migration and inflammogenesis.

We thank the reviewers for this suggestion. We now show that blocking GM-CSF does return the cryptopatch balance of entrance and exit to steady-state levels (Figure 5). This solidifies the role of GM-CSF in the earliest stages of inflammation. This data with the increase of IL-23R at 6 hours after anti-CD40 treatment (Figure 4—figure supplement 1) supports an early pro-inflammatory loop between IL-23 producing myeloid cells and GM-CSF producing ILCs.